# Assessment of Knowledge, Perception, Experience and Phobia toward Corticosteroids Use among the General Public in the Era of COVID-19: A Multinational Study

**DOI:** 10.3390/healthcare11020255

**Published:** 2023-01-13

**Authors:** Muna Barakat, Mohamed Hassan Elnaem, Amani Al-Rawashdeh, Bayan Othman, Sarah Ibrahim, Doaa H. Abdelaziz, Anas O. Alshweiki, Zelal Kharaba, Diana Malaeb, Nabeel Kashan Syed, Abdulqadir J. Nashwan, Mohammed Fathelrahman Adam, Reem Alzayer, Mohammad Saleh Albarbandi, Rana K. Abu-Farha, Malik Sallam, Yasmeen Barakat, Noha O. Mansour

**Affiliations:** 1Department of Clinical Pharmacy and Therapeutics, Faculty of Pharmacy, Applied Science Private University, Amman 11931, Jordan; 2MEU Research Unit, Middle East University, Amman 11831, Jordan; 3School of Pharmaceutical Sciences, Universiti Sains Malaysia, George Town 11800, Malaysia; 4Faculty of Pharmacy, Amman Arab University, Amman 11953, Jordan; 5Department of Pharmaceutical Sciences and Pharmaceutics, Faculty of Pharmacy, Applied Science Private University, Amman 11931, Jordan; 6Department of Biopharmaceutics and Clinical Pharmacy, School of Pharmacy, The University of Jordan, Amman 11942, Jordan; 7Pharmacy Practice and Clinical Pharmacy Department, Faculty of Pharmacy, Future University in Egypt, Cairo 11835, Egypt; 8Department of Clinical Pharmacy, the National Hepatology and Tropical Medicine Research Institute, Cairo 11835, Egypt; 9Department of Clinical Pharmacy, College of Pharmacy, Al Ain University, Abu Dhabi 112612, United Arab Emirates; 10AAU Health and Biomedical Research Center, Al Ain University, Abu Dhabi 112612, United Arab Emirates; 11Faculty of Medical Sciences, Newcastle University, Newcastle Upon Type NE2 4HH, UK; 12College of Pharmacy, Gulf Medical University, Ajman P.O. Box 4184, United Arab Emirates; 13School of Pharmacy, Lebanese International University, Beirut 1083, Lebanon; 14Pharmacy Practice Research Unit, Department of Pharmacy Practice, College of Pharmacy, Jazan University, Gizan 45142, Saudi Arabia; 15Department of Nursing Education & Practice Development, Hazm Mebaireek General Hospital (HMGH), Hamad Medical Corporation (HMC), Doha 3050, Qatar; 16Faculty of Nursing, University of Calgary in Qatar (UCQ), Doha 23133, Qatar; 17Faculty of Pharmacy, University of Science and Technology, Omdurman 14411, Sudan; 18Clinical pharmacy Practice, Department of pharmacy, Mohammed Al-Mana College for Medical Sciences, Dammam 34222, Saudi Arabia; 19Department of Neurosurgery, Ibn Al-Nafees Hospital, Damascus G8W4+MQW, Syria; 20Department of Neurosurgery, Damascus Hospital, Damascus G72W+25C, Syria; 21Department of Pathology, Microbiology and Forensic Medicine, School of Medicine, The University of Jordan, Amman 11942, Jordan; 22Department of Clinical Laboratories and Forensic Medicine, Jordan University Hospital, Amman 11942, Jordan; 23Department of Translational Medicine, Faculty of Medicine, Lund University, 22184 Malmö, Sweden; 24Clinical Pharmacy and Pharmacy Practice Department, Faculty of Pharmacy, Mansoura University, Mansoura 35516, Egypt

**Keywords:** corticosteroids, corticophobia, COVID-19, knowledge, perception

## Abstract

Background: Corticosteroids play a significant role in managing the vast majority of inflammatory and immunologic conditions. To date, population-based studies on knowledge and attitudes concerning corticosteroids are scarce. This study aims to comprehensively assess knowledge, perception, experience and phobia toward corticosteroid use among the general population in the era of COVID-19. Methods: A cross-sectional self-administrated questionnaire was used to collect the data from 6 countries. Knowledge and corticophobia scores, descriptive statistics and logistic regression were computed. Results: A total of 2354 participants were enrolled in this study; the majority were females (61.6%) with an average age of 30. Around 61.9% had been infected previously with COVID-19, and about one-third of the participants had experience with corticosteroid use. The mean knowledge score was relatively satisfactory (8.7 ± 4.5 out of 14), and Corticophobia ranked a high score in all countries. Age, female gender, and history of COVID-19 were positively correlated with developing corticophobia. Conclusion: Our study highlights that the general knowledge about steroids was satisfactory. However, the phobia toward its use upon indication is high. Therefore, enhancing awareness and providing essential counseling regarding the rational use of corticosteroids may reduce corticophobia.

## 1. Introduction

COVID-19 is a rapidly spreading infectious disease caused by severe acute respiratory syndrome coronavirus 2 (SARS-CoV-2). It is characterized by a hyperinflammatory response; severe cases develop acute respiratory distress syndrome (ARDS), the most severe form of acute respiratory failure. Although many Arab countries applied early serious preventive measures to control the infection of COVID-19, the total number of confirmed COVID-19 cases in the Arab world ranked fourth among the top 15 most affected countries [1]. This increasing trend continued despite the progress in clinical trials concerning exploring different therapies to combat COVID-19 [2]; indicating that native factors is linked to COVID-19 infection and severity [3].

Corticosteroids play a significant role in managing the vast majority of inflammatory and immunologic disorders [4]. They gained exceptional global attention after the recent emergent evidence which linked corticosteroid use in COVID-19 patients with positive clinical outcomes [5]. Corticosteroid use might be associated with risks ranging from mild adverse drug reactions to serious hazards such as cardiovascular disorders and/or immunosuppression [4,6]. These significant health risks are most likely linked to the prolonged systemic use of high doses [4]. Optimal clinical and safety outcomes are positively correlated with patients’ knowledge levels. Therefore, determining community understanding of medications and appropriate utilization is generally a cornerstone in achieving an optimal therapeutic outcome [7,8]. Considering that corticosteroids are easily accessible and affordable treatment options in developed countries, hence evaluation of the public awareness of these agents in those countries is of particular significance [9].

Concerns regarding corticosteroid use, broadly known as “corticophobia” [10]. Corticophobia is defined as “Exaggerated concerns, fears, worries, anxiety, doubts, reservations, reluctance or skepticism regarding corticosteroid use in patients, their caregivers, or health care professionals” [11]. It is a universal problem, particularly in developed countries where public awareness about medication is hampered [12,13,14]. Corticophobia-related non-adherence has been estimated to be present in up to 58% of patients, emphasizing the importance of this issue [15]. This non-adherence may hamper disease control and heavily burdens healthcare resources [10,15].

Understanding basic principles about important drug categories, like corticosteroids, is essential to health literacy. This knowledge enables individuals to comprehend better the information that they are likely to receive about those drugs [16]. Studies investigating medication adherence in Arab patients generally have reported low levels. For example, a study undertaken with Saudi Arabian patients reported a lower adherence rate of 22.5%. Various factors such as economic [17] and cultural attributes [18], health beliefs might have a special influence on adherence to medications among those people [17,19]. Compared to Western countries the important forementioned medication adherence factors hugely differ from Arab countries [20].

Population-based studies on knowledge and attitudes concerning corticosteroids are scarce, and the results are strongly limited by the small sample size and investigation of a single dosage form [21]. Another notable limitation is focusing on specific populations such as healthcare professionals [22], caregivers [23,24,25,26], and/or special patient groups [27,28]. The results of those studies reported limited knowledge of corticosteroids and their optimal usage. So far, no multinational study has investigated the public understanding of corticosteroid use. Describing the prevalence of corticophobia and identifying its underlying factors have not been previously investigated in developed countries. This multinational study aims to assess knowledge, awareness, and attitudes and examine the phobia surrounding corticosteroid use among the general public from different Arab countries in the era of COVID-19.

## 2. Materials and Methods

This descriptive cross-sectional study was conducted between March and July 2022 among the general population from 6 Arab countries including Egypt, Iraq, Jordan, Saudi Arabia, Sudan, and Syria. A self-administered, online questionnaire was delivered to the participants in two languages (English and Arabic) using Google Forms. In addition, 89 participants responded to the questionnaire from other Arab countries such as Kuwait, Lebanon, Palestine, Qatar, and Yemen, which were categorized in the study as “others”. Regarding the inclusion criteria, the eligible participants include any adult (age > 18 years) who was able to read and respond to the questionnaire. The data recruitment process was accomplished through the co-investigators in each country.

### 2.1. Sample Size

This was a multinational study with high variability in the number of populations. Accordingly, based on Tabachnick and Fidell’s recommendation for sample size calculation in analysis, 5–20 subjects per predictor are suggested to be preferable [29]. Based on the number of independent variables levels used in this study (*n* = 9) and the number of 10 subjects per predictor level, a minimum sample size of 90 or higher from each country was considered suitable.

### 2.2. Ethical Statement

Participation in this study was voluntary and anonymous. Informed consent was obtained from each participant. A written participant consent statement, “Your participation in completing this questionnaire is highly appreciated”, was given to the participants at the beginning of the survey. Consent approval was achieved if the participants were willing to proceed with the survey.

### 2.3. Study Tool

The study questionnaire was developed after reviewing related validated surveys in the literature [23,30,31] and was designed using the general principles of good survey design [32]. Several sources were used to generate a pool of questions considered relevant to the objectives of the study. The questionnaire contained 19 multiple-choice questions, designed to be completed within 7–10 min. It was comprised of four main parts: First part (10 questions to cover the general sociodemographic details and COVID-19 history of the participants). The second part (5 questions to gather information about participants’ experience with corticosteroid use). The third part (3 questions to assess the participants’ knowledge about corticosteroids). The last part (one question includes 12 statements to examine the level of corticophobia among participants).

Initially, the tool was developed in English and evaluated by five academics in the pharmacy field for content and face validity. Then, it was translated into Arabic by two independent academic translators using translation and back-translation techniques. For the sake of clarity and understandability of the questionnaire’s items among the study-eligible population, it was piloted on 30 participants from various backgrounds (academic and professional), and linguistic refinements were made as needed based on the feedback from the pilot-testing phase. The pilot responses were not included in the final analyses. The final version of the questionnaire was distributed through online media, mainly social media, e.g., Facebook [33], WhatsApp [34], and LinkedIn [35] messages using the snowball-convenience sampling method.

The total knowledge score was calculated by giving the participants a list of 14 knowledge items. Each item had two response choices, either yes or no, which were scored based on whether they were correct (score of 1) or incorrect (score of 0); thus, the overall knowledge score ranges between (0 and 14). There was no cut-off score value that identified the acceptable level of knowledge; however, this study summed up the total correct answers and considered a good level of knowledge above 50%. In addition, a higher score in each category represents a higher knowledge level.

Regarding the corticophobia score, the responses for the listed statements were recorded using a 5-point Likert scale (Strongly disagree = 1, disagree = 2, neutral = 3, agree = 4, and strongly agree = 5). Through Cronbach’s alphas, the reliability of the scale was determined, and it showed that the alpha was 0.83, which indicated that the items would form a scale of high internal consistency. The results of the 5-point Likert type scale were analyzed and ranked between 1–5 have been divided into three ranks: a low score is between 1.00–1.66, a moderate score between 1.67–3.32, and a high score between 3.33 and 5.00.

### 2.4. Statistical Analysis

Study data was analyzed using the 24th version of the statistical package for social science (SPSS^®^). The mean ± standard deviation (S.D.) and frequency (or percentages) were used for continuous and categorical variables, respectively. The Chi-square and Kruskal–Wallis tests were used to identify a significant difference in sociodemographic variables between countries, and for frequencies < 5, Fisher’s exact test was used. Binary logistic regression analyses were used to test the predictors for corticophobia. Univariate logistic regression was carried out to initially screen the independent variables associated with high scores (3.33–5.00) on the Likert scale for corticophobia, including age, gender, major, education level, residential area, presence of chronic diseases, history of COVID-19 infection, experience with corticosteroid use, experience with side effects from corticosteroids and knowledge score about corticosteroids. Using univariate regression analysis, predictors with *p*-value < 0.25 were entered into multivariable regression analysis. Predictors were selected after checking their independence, where Spearman’s correlation coefficient (r) of less than 0.4 indicates the absence of multicollinearity between the independent predictors in multivariable regression analysis. The multivariable regression analysis identified independent variables associated with high scores. Statistical significance was set at a *p*-value of test ≤ 0.05.

## 3. Results

### 3.1. Sociodemographic Characteristics

A total of 2354 participants from six Arab countries completed the study survey. The majority of the participants were females (61.6%), with an average age of 30 years. Almost two-thirds of the participants were bachelor’s degree holders (63.5%), and a majority reported living in urban areas (83.6%). Approximately one-third of the sample (29.3%) were students, (29.1%) worked in the healthcare sector, and (18.7%) were unemployed or retired. Regarding the health status of the participants, the majority did not report having any chronic disorder (81.7%). Of those with chronic conditions, the most prevalent were cardiovascular diseases (9.6%) or obesity/overweight (8.4%). Table 1 shows the sociodemographic characteristics and chronic disease status of study participants.

### 3.2. History of COVID-19 Infection

Around (61.9%) of the total study subjects had been infected previously with COVID-19, where 21.5% rated that the infection was mild. Across all participating countries, almost a quarter of the involved individuals have been infected once only with COVID-19 (26%). Exactly one-third of the total sample admitted receiving the COVID-19 vaccine (33.3%). Table 2 shows the history of COVID-19 infection of the participants in this study.

### 3.3. Experience with Corticosteroid Use

Around (31.6%) of the subjects reported using corticosteroids themselves or having relatives or friends who do. Among those who had previous exposure to corticosteroid usage, the majority (36.7%) had continuously used the corticosteroid medication for less than 7 days. The most commonly utilized dosage form (62.1%) was the tablet. More than half of the participants (51.5%) used corticosteroids for the treatment of dermatologic disorders, 26.1% for infection with COVID-19, and 41% for the treatment of respiratory disorders (e.g., Asthma, chronic obstructive pulmonary disease (COPD). Figure 1 shows what the participants said when asked about their exposure to corticosteroids for any reason.

The most commonly reported side effects among study respondents were increased appetite leading to weight gain (45%) followed by mood changes, swings, and depression (34.1%). Also, more than a quarter of the participants who used corticosteroids reported acne or easily bruised thin skin (27.8% and 26.1%), respectively. Table 3 shows reported experiences with corticosteroid use among the participants from different countries.

### 3.4. Knowledge about Corticosteroid

Participants from different countries showed relatively satisfactory knowledge (mean > 7 out of 14) about corticosteroids. Participants scored a total of (8.7 ± 4.5) out of (14) where each correct answer scored 1 and 0 for the incorrect answer. The highest knowledge score was reported from Egypt and Iraq (10.5), while the lowest was from Saudi Arabia (7.5). Table 4 shows participants’ frequencies (percentages) of correct answers to the 14 knowledge questions about corticosteroids from the different countries.

Regarding the information sources from which most participants generated their knowledge about corticosteroids, social media platforms appeared to be one of the most utilized sources to acquire information about corticosteroids (66.8%), followed by books (59.6%). In countries like Egypt, Jordan, and Sudan, healthcare workers were the least preferred source of information accessibility about corticosteroids. Figure 2 demonstrates the different sources of information used by the study participants to gain knowledge about corticosteroids.

### 3.5. Corticophobia among the Participants

Fear of corticosteroid use (corticophobia) scores were assessed among the study participants from different countries. Overall, corticophobia scores appeared high (3.33–5.00) on the Likert scale among all participants on all assessed corticophobia aspects, as shown in Table 5. Participants expressed their fear of corticosteroid-related side effects, change of dosage form, or use of corticosteroids as a drug of choice.

Regression results at 95% confidence interval revealed that a high Corticophobia score was positively correlated with female gender, age, and history of COVID-19 infection. In addition, participants’ educational level and knowledge scores were negatively correlated with developing corticophobia. Table 6 presents significant predictors of high corticophobia scores among the study participants.

## 4. Discussion

Corticosteroids can be prescribed as powerful anti-inflammatory therapeutic medications used for the treatment of a wide variety of disorders. Nevertheless, they are associated with some side effects that are mostly related to the dose used and duration of treatment. During the COVID-19 pandemic, many reported behaviors were reported regarding medications use among different countries including Middle East [36,37,38]. Furthermore, a debate on the use of corticosteroids was going on until several results of studies led to amendments recommending corticosteroids among the COVID-19 treatment guidelines in certain cases [9,39,40]. This study aimed to comprehensively assess the public knowledge and fears about using corticosteroids among different Arab countries in the era of COVID-19.

Previous studies were conducted to assess knowledge and awareness regarding corticosteroids [41,42,43,44]. For example, in India, knowledge and awareness regarding prescription steroids were very low; around 83% of participants were unfamiliar with its related information [41]. Contrary to our study findings, although less than a third of the sample reported previous usage of corticosteroids (31.6%), relatively good knowledge scores among participants were found. This might be attributed to the educational level of the participants, as more than half were bachelor’s degree holders (63.5%), and the majority reported living in urban areas (83.6%) rather than rural areas. It is well known that sociodemographic characteristics, primarily educational level, are strongly correlated to health [45,46,47], which might have contributed to this finding. Similar findings were reported by a recent study In India, assessing the knowledge and awareness among people visiting the general outpatient department of a tertiary care hospital [41].

On the other hand, low levels of knowledge regarding corticosteroid side effects among study participants were found. A nationwide survey in south Korea reported similar findings on the general public knowledge regarding topical corticosteroids. Fear of side effects is a major limiting factor for the use of steroids. Hence, the role of health care providers is crucial in appropriately counseling patients on the correct use of corticosteroids and preventing hesitation, misuse, and abuse of corticosteroids.

The present study showed that most participants generated their knowledge about corticosteroids mainly from social media platforms (66.8%). Internet accessibility, mainly social media, is reported to be one of the most feasible and accessible sources of health information nowadays [48,49,50]. Studies have reported several uses of social media by healthcare providers, including disseminating health information and health promotion [50,51,52,53]. Despite the feasibility and accessibility of these platforms, these sources sometimes are not validated for medical information, which could lead to medication misuse and potential side effects [54,55]. Additionally, in our study, healthcare workers in countries like Egypt, Jordan, and Sudan, were reported to be the least preferred choice for information about corticosteroids among participants. Alarming low selection rates of healthcare professionals shed light on the possible misuse of drugs, which could lead to serious side effects. Therefore, the healthcare provider’s role is to appropriately counsel the patients on the correct use of corticosteroids and prevent hesitation, misuse, and abuse of corticosteroids. Hence, patients trust the healthcare providers and have an essential input in their perception of medication use, including phobia [56].

During the COVID-19 pandemic, social media platforms have been crucial in transmitting health information between people [57]. A previous study from the Middle East during the pandemic reported similar findings on social media platforms being the most common source of information among people [58]. Also, during the pandemic, the use of corticosteroids (Dexamethasone) as a preventive medication increased due to the media blitz among patients [59,60]. In addition, in dermatology, an increase in oral and topical corticosteroid usage as self-medications during COVID-19 was reported as well [61]. In the present study, although (61.9%) of participants had been infected with COVID-19, only (31.6%) of this sample reported previous use of corticosteroids. Among those who had prior exposure to corticosteroid usage, 36.7% had continuously used a corticosteroid medication for less than seven days. In addition, corticophobia scores were found to be high (3.33–5.00) on the Likert scale among the study participants from different countries. All these findings indicate the presence of steroid phobia or “corticophobia” among patients, defined as ambiguous negative feelings and beliefs about corticosteroid usage [10].

Steroid phobia is a critical hurdle that affects the acceptability of treatment, and adherence to it eventually worsens disease progression [62]. Several studies were conducted to assess corticophobia among patients [28,63,64]. In this study, participants expressed fear of corticosteroid-related side effects, including weight gain (45%), mood changes, mood swings, and depression (34.1%). This emphasizes the critical role of health providers in addressing steroid phobia and assessing adherence among patients.

Several studies investigated associations with developing corticosteroid phobia. In our study, age and gender were positively correlated with developing corticophobia. Several studies reported similar findings, as females were more prone to convey a steroid phobia, especially if they have experienced any side effects from corticosteroid use [28,65,66]. On the other hand, divergent findings were reported regarding age, specific patients below 60 years [65,66].

Participants’ educational level and knowledge scores were significantly negatively correlated with developing corticophobia. As a person would expect, education usually improves people’s general knowledge and awareness, which might explain this association of steroid phobia with educational level and knowledge scores. Nevertheless, another study highlighted that steroid phobia might be associated with having a higher university degree [66].

Several factors might contribute to corticophobia among people. These factors might include the fear of side effects, lack of education, lack of counseling, and misinformation from social media. The fears of side effects such as increased appetite and weight gain, as reported by this study and others, indicate that fear of adverse effects and lack of appropriate knowledge about steroid treatment are the most important reasons for developing corticophobia [67,68]. Therefore, strategies to assess corticophobia among users are essential as it affects medication adherence and treatment outcomes. Health care providers’ role is crucial in properly educating and counseling patients on the usage of corticosteroids. However, it is not enough alone as, interestingly, improving patients’ knowledge was found ineffective at reducing fear or improving adherence [69]. Building a trusting relationship between healthcare providers and patients is also essential, leading to better communication, better delivery of information and education, and eventually, an efficient reduction in steroid-related stress in patients [70].

Several limitations can be identified in this study. First, the representation of each participating population could be compromised, as the study tool warrants computer literacy, internet availability, an enhanced level of education to access and complete the online survey, which might affect the generalizability of our study findings. Second, information bias related to the accessibility of resources on demand can compromise response credibility. Third, selection bias related to the data collection (snowball technique) might be an issue, with no random selection warranted. Residual confounding bias could arise from possible un-measured variables or responses to variables directly or indirectly related to corticosteroids. Moreover, an online survey instead of a face-to-face meeting poses reliability and authenticity risks to the study data. The online survey included country-specific questions for each population to complete, with a full description of the target population and inclusion criteria in the title and the invitation message. Considering the restriction measures during the COVID-19 pandemic, such a methodology was the best option.

## 5. Conclusions

With corticosteroids being used extensively, it is imperative to control the use of these medications in the proper way. In our study, while the general level of knowledge was satisfactory about steroid usage, indications, and dosage regimen, there is still a need to strengthen the knowledge and understanding regarding their possible side effects at a deeper level. In addition, this study revealed that the corticophobia level was high; therefore, enhancing awareness and providing essential counseling regarding the rational use of corticosteroids may reduce corticophobia.

## Figures and Tables

**Figure 1 healthcare-11-00255-f001:**
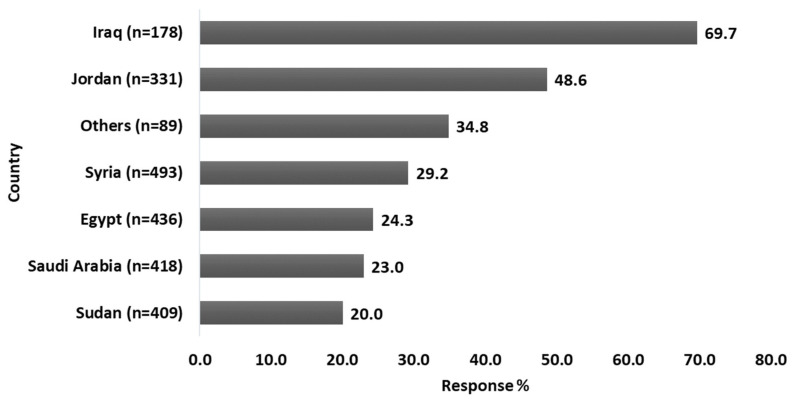
Participants’ response rate upon asking them if they ever (or any of their relatives/friends) use corticosteroids for any reason. Among the total sample size, only 744 (31.6%) participants had an experience with corticosteroid usage.

**Figure 2 healthcare-11-00255-f002:**
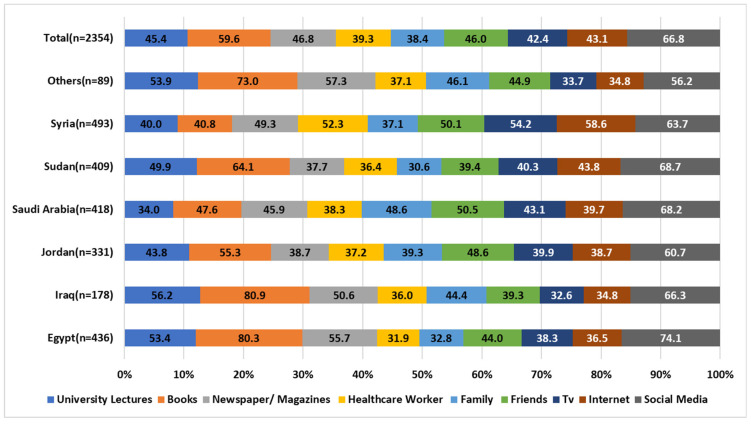
Sources of information about corticosteroids. Participants were able to choose more than one option.

**Table 1 healthcare-11-00255-t001:** Sociodemographic characteristics of study participants (*n* = 2354).

Variable	Country								
	Egypt (*n* = 436)	Iraq (*n* = 178)	Jordan (*n* = 331)	Saudi Arabia (*n* = 418)	Sudan (*n* = 409)	Syria (*n* = 493)	Others (*n* = 89)	Total (*n* = 2354)	*p*-Value *
**Age, years, median (IQR)**	25.0 (58.0)	26.0 (49.0)	31.0 (56.0)	34.0 (57.0)	24.0 (53)	39.0 (57.0)	32.0 (47.0)	30.0 (58.0)	<0.001 ^$^
**Gender**									
Male, *n* (%)	135 (31.0)	53 (29.8)	119 (36.0)	127 (30.4)	119 (29.1)	315 (63.9)	36 (40.4)	904 (38.4)	<0.001
Female, *n* (%)	301 (69.0)	125 (70.2)	212 (64.0)	291 (69.6)	290 (70.9)	178 (36.1)	53 (59.6)	1450 (61.6)	
**Educational level**									<0.001
School level, *n* (%)	38 (8.7)	15 (8.4)	19 (5.7)	57 (13.6)	24 (5.9)	116 (23.5)	5 (5.6)	274 (11.6)	
Diploma, *n* (%)	20 (4.6)	10 (5.6)	31 (9.4)	45 (10.8)	20 (4.9)	124 (25.2)	5 (5.6)	255 (10.8)	
Bachelor, *n* (%)	318 (72.9)	112 (62.9)	223 (67.4)	260 (62.2)	340 (83.1)	181 (36.7)	60 (67.4)	1494 (63.5)	
Postgraduates, *n* (%)	60 (13.8)	41 (23.0)	58 (17.5)	56 (13.4)	25 (6.1)	72 (14.6)	19 (21.3)	331 (14.1)	
**Residential area**									
Urban, *n* (%)	302 (69.3)	135 (75.8)	306 (92.4)	376 (90.0)	368 (90.0)	403 (81.7)	77 (86.5)	1967 (83.6)	<0.001
Rural, *n* (%)	134 (30.7)	43 (24.2)	25 (7.6)	42 (10.0)	41 (10.0)	90 (18.3)	12 (13.5)	387 (16.4)	
**Major**									
Student, *n* (%)	187 (42.9)	56 (31.5)	68 (20.5)	81 (19.4)	231 (56.5)	52 (10.5)	14 (15.7)	689 (29.3)	<0.001
Health sector, *n* (%)	126 (28.9)	76 (42.7)	102 (30.8)	124 (29.7)	81 (19.8)	141 (28.6)	34 (38.2)	684 (29.1)	
Non-Health sector, *n* (%)	49 (11.2)	32 (18.0)	86 (26.0)	107 (25.6)	56 (13.7)	188 (38.1)	22 (24.7)	540 (22.9)	
Unemployed/retired, *n* (%)	74 (17.0)	14 (7.9)	75 (22.7)	106 (25.4)	41 (10.0)	112 (22.7)	19 (21.3)	441 (18.7)	
**Do you have any chronic diseases?**									
No, *n* (%)	382 (87.6)	153 (86.0)	272 (82.2)	306 (73.2)	339 (82.9)	401 (81.3)	71 (79.8)	1924 (81.7)	
Yes, *n* (%)	54 (12.4)	25 (14.0)	59 (17.8)	112 (26.8)	70 (17.1)	92 (18.7)	18 (20.2)	430 (18.3)	<0.001
**If yes, which of the following could apply? n (%)**									
Hypertension (High blood pressure)	9 (2.1)	6 (3.4)	18 (5.4)	27 (6.5)	11 (2.7)	61 (12.4)	6 (6.7)	138 (5.9)	<0.001
Diabetes mellitus	34 (7.8)	25 (14.0)	13 (3.9)	27 (6.5)	9 (2.2)	29 (5.9)	18 (20.2)	155 (6.6)	<0.001
Obesity/overweight	34 (7.8)	17 (9.6)	32 (9.7)	40 (9.6)	23 (5.6)	47 (9.5)	5 (5.6)	198 (8.4)	<0.001
Cardiovascular diseases (e.g., Stroke)	54 (12.4)	20 (11.2)	11 (3.3)	66 (15.8)	41 (10.0)	17 (3.4)	18 (20.2)	227 (9.6)	<0.001
Kidney disease	5 (1.1)	5 (2.8)	6 (1.8)	11 (2.6)	11 (2.7)	18 (3.7)	5 (5.6)	61 (2.6)	<0.001
Osteoporosis	10 (2.3)	12 (6.7)	6 (1.8)	12 (2.9)	8 (2.0)	62 (12.6)	11 (12.4)	121 (5.1)	<0.001
Depression	35 (8.0)	10 (5.6)	17 (5.1)	23 (5.5)	20 (4.9)	50 (10.1)	5 (5.6)	160 (6.8)	<0.001
Rheumatoid Arthritis	9 (2.1)	6 (3.4)	10 (3.0)	21 (5.0)	9 (2.2)	60 (12.2)	5 (5.6)	120 (5.1)	<0.001
Immune disorder	13 (3.0)	6 (3.4)	14 (4.2)	14 (3.3)	9 (2.2)	20 (4.1)	6 (6.7)	82 (3.5)	<0.001
Respiratory disease (e.g., Asthma)	19 (4.4)	8 (4.5)	14 (4.2)	30 (7.2)	35 (8.6)	30 (6.1)	13 (14.6)	149 (6.3)	<0.001
Dermatological disorder (e.g., acne, psoriasis)	36 (8.3)	15 (8.4)	15 (4.5)	37 (8.9)	34 (8.3)	32 (6.5)	18 (20.2)	187 (7.9)	<0.001
Others	31 (7.1)	12 (6.7)	25 (7.6)	42 (10.0)	26 (6.4)	27 (5.5)	5 (5.6)	168 (7.1)	<0.001

* Significance measure at *p* < 0.05, chi-square analysis. ^$^ Kruskal–Wallis H test. Note: each percentage was calculated based on the frequency of the participating country.

**Table 2 healthcare-11-00255-t002:** Participants reported history of COVID-19 infection.

	Residential Country							
Question	*n* (%)							
	Egypt (*n* = 436)	Iraq (*n* = 178)	Jordan (*n* = 331)	Saudi Arabia (*n* = 418)	Sudan (*n* = 409)	Syria (*n* = 493)	Others (*n* = 89)	Total (*n* = 2354)
**Have you been infected with COVID-19?**								
○ No	289 (66.3)	65 (36.5)	129 (39.0)	267 (63.9)	332 (81.2)	329 (66.7)	46 (51.7)	1457 (61.9)
○ Yes	147 (33.7)	113 (63.5)	202 (61.0)	151 (36.1)	77 (18.8)	164 (33.3)	43 (48.3)	897 (38.1)
**How did you rate the severity**								
○ Critical (have been admitted to intensive care unit (ICU))	0 (0.0)	1 (0.6)	0 (0.0)	4 (1.0)	0 (0.0)	1 (0.2)	0 (0.0)	6 (0.3)
○ Mild (exhibit a variety of signs and symptoms, but without shortness of breath or abnormal imaging)	84 (19.3)	58 (32.6)	111 (33.5)	86 (20.6)	53 (13.0)	92 (18.7)	22 (24.7)	506 (21.5)
○ Moderate (had evidence of lower respiratory disease during clinical assessment or imaging, with SpO_2_ ≥ 94% at room air)	60 (13.8)	52 (29.2)	82 (24.8)	47 (11.2)	23 (5.6)	62 (12.6)	18 (20.2)	344 (14.6)
○ Severe (have SpO_2_ < 94% on room air and administered oxygen therapy)	3 (0.7)	2 (1.1)	9 (2.7)	14 (3.3)	1 (0.2)	9 (1.8)	3 (3.4)	41 (1.7)
**How many times have you been infected with COVID-19?**								
○ 1	88 (20.2)	62 (34.8)	138 (41.7)	121 (28.9)	52 (12.7)	118 (23.9)	34 (38.2)	613 (26.0)
○ 2	45 (10.3)	35 (19.7)	57 (17.2)	28 (6.7)	20 (4.9)	33 (6.7)	8 (9.0)	226 (9.6)
○ 3 or more	14 (3.2)	16 (9.0)	7 (2.1)	2 (0.5)	5 (1.2)	13 (2.6)	1 (1.1)	58 (2.5)
**Did you get your COVID-19 vaccine?**								
○ No	9 (2.1)	15 (8.4)	8 (2.4)	4 (1.0)	42 (10.3)	27 (5.5)	9 (10.1)	114 (4.8)
○ Yes	138 (31.7)	98 (55.1)	194 (58.6)	147 (35.2)	35 (8.6)	137 (27.8)	34 (38.2)	783 (33.3)

Note: each percentage was calculated based on the frequency of the participating country.

**Table 3 healthcare-11-00255-t003:** Participants’ reported experience with corticosteroid usage in different countries (total *n* = 744).

	Residential Country *n* (%)							
Question	Egypt	Iraq	Jordan	Saudi Arabia	Sudan	Syria	Others	Total
**If your answer was yes, how long did you (or he/she) use it?**								
<1 week	27 (25.5)	74 (59.7)	61 (37.9)	20 (20.8)	25 (30.5)	55 (38.2)	11 (35.5)	273 (36.7)
>3 months	10 (9.4)	36 (29.0)	42 (26.1)	35 (36.5)	23 (28.0)	37 (25.7)	7 (22.6)	190 (25.5)
1–3 months	14 (13.2)	9 (7.3)	20 (12.4)	12 (12.5)	14 (17.1)	18 (12.5)	3 (9.7)	90 (12.1)
1–4 weeks	55 (51.9)	5 (4.0)	38 (23.6)	29 (30.2)	20 (24.4)	34 (23.6)	10 (32.3)	191 (25.7)
**What was the used dosage form? More than one option is/are allowed**								
Topical (e.g., cream or ointment)	50 (47.2)	63 (50.8)	90 (55.9)	51 (53.1)	41 (50.0)	61 (42.4)	19 (61.3)	375 (50.4)
Inhaler or nebulizer	40 (37.7)	37 (29.8)	65 (40.4)	29 (30.2)	26 (31.7)	47 (32.6)	10 (32.3)	254 (34.1)
Tablets	71 (67.0)	75 (60.5)	105 (65.2)	48 (50.0)	52 (63.4)	86 (59.7)	25 (80.6)	462 (62.1)
Injection	42 (39.6)	64 (51.6)	67 (41.6)	21 (21.9)	16 (19.5)	50 (34.7)	13 (41.9)	273 (36.7)
Drops (e.g., eye drops)	24 (22.6)	29 (23.4)	36 (22.4)	7 (7.3)	18 (22.0)	22 (15.3)	9 (29.0)	145 (19.5)
**What was the main indication for corticosteroids use? More than one option is/are allowed**								
Respiratory disease (e.g., Asthma, COPD)	45 (42.5)	41 (33.1)	75 (46.6)	30 (31.3)	37 (45.1)	61 (42.4)	16 (51.6)	305 (41.0)
COVID-19	45 (42.5)	47 (37.9)	51 (31.7)	11 (11.5)	13 (15.9)	18 (12.5)	9 (29.0)	194 (26.1)
Dermatological disorders (e.g., Eczema)	41 (38.7)	52 (41.9)	90 (55.9)	54 (56.3)	40 (48.8)	88 (61.1)	18 (58.1)	383 (51.5)
Joint or Rheumatological disorders	21 (19.8)	31 (25.0)	33 (20.5)	16 (16.7)	21 (25.6)	64 (44.4)	7 (22.6)	193 (25.9)
GIT Immunological disorders (e.g., Crohn’s disease, ulcerative colitis)	9 (8.5)	11 (8.9)	24 (14.9)	10 (10.4)	13 (15.9)	17 (11.8)	3 (9.7)	87 (11.7)
Systemic Immunological disorders (e.g., Multiple Sclerosis)	9 (8.5)	10 (8.1)	22 (13.7)	7 (7.3)	12 (14.6)	24 (16.7)	4 (12.9)	88 (11.8)
Others	25 (23.6)	39 (31.5)	39 (24.2)	18 (18.8)	25 (30.5)	50 (34.7)	12 (38.7)	208 (28.0)
**Did you suffer from any of the following side effects following Corticosteroid usage? More than one option is/are allowed**								
Increased Appetite—Potentially leading to weight gain	32 (30.2)	51 (41.1)	74 (46.0)	34 (35.4)	29 (35.4)	97 (67.4)	18 (58.1)	335 (45.0)
Acne	18 (17.0)	31 (25.0)	29 (18.0)	15 (15.6)	22 (26.8)	84 (58.3)	8 (25.8)	207 (27.8)
Thinned skin that bruises easily	21 (19.8)	24 (19.4)	33 (20.5)	28 (29.2)	19 (23.2)	60 (41.7)	9 (29.0)	194 (26.1)
Increased risk of infections	20 (18.9)	37 (29.8)	24 (14.9)	14 (14.6)	17 (20.7)	38 (26.4)	7 (22.6)	157 (21.1)
Mood changes, mood swings, and Depression	37 (34.9)	47 (37.9)	66 (41.0)	28 (29.2)	31 (37.8)	32 (22.2)	13 (41.9)	254 (34.1)
Hyperglycemia or Diabetes	6 (5.7)	10 (8.1)	12 (7.5)	8 (8.3)	6 (7.3)	15 (10.4)	2 (6.5)	59 (7.9)
High blood pressure	12 (11.3)	24 (19.4)	18 (11.2)	11 (11.5)	14 (17.1)	17 (11.8)	6 (19.4)	102 (13.7)
Osteoporosis (Weak and brittle bones)	11 (10.4)	23 (18.5)	19 (11.8)	10 (10.4)	10 (12.2)	66 (45.8)	6 (19.4)	145 (19.5)
Others	9 (8.5)	28 (22.6)	28 (17.4)	10 (10.4)	17 (20.7)	31 (21.5)	6 (19.4)	129 (17.3)

Note: each percentage was calculated based on the frequency of the participants who had an experience with corticosteroids; Egypt (*n* = 106), Iraq (*n* = 124), Jordan (*n* = 161), Saudi Arabia (*n* = 96), Sudan (*n* = 82), Syria (*n* = 144), Others (*n* = 31), Total (*n* = 744).

**Table 4 healthcare-11-00255-t004:** Participants’ frequencies (percentages) of correct answers to the knowledge questions about corticosteroids from the different countries.

	Residential Country *n* (%)							
Statements	Egypt (*n* = 436)	Iraq (*n* = 178)	Jordan (*n* = 331)	Saudi Arabia (*n* = 418)	Sudan (*n* = 409)	Syria (*n* = 493)	Others (*n* = 89)	Total (*n* = 2354)
**1.** **Corticosteroids, often known as steroids, are anti-inflammatory medicine.**	377 (86.5)	150 (84.3)	210 (63.4)	258 (61.7)	265 (64.8)	315 (63.9)	71 (79.8)	1646 (69.9)
**2.** **Corticosteroids are man-made hormones normally produced by the adrenal glands.**	359 (82.3)	138 (77.5)	196 (59.2)	201 (48.1)	233 (57.0)	301 (61.1)	48 (53.9)	1476 (62.7)
**3.** **Corticosteroids are mainly used to reduce inflammation and suppress the immune system.**	341 (78.2)	140 (78.7)	221 (66.8)	239 (57.2)	276 (67.5)	302 (61.3)	62 (69.7)	1581 (67.2)
**4.** **Corticosteroids are used to treat various health conditions (e.g., asthma, eczema, COVID-19. etc.).**	330 (75.7)	134 (75.3)	224 (67.7)	262 (62.7)	253 (61.9)	260 (52.7)	60 (67.4)	1523 (64.7)
**5.** **Prolonged steroid treatment at high doses—particularly with steroid tablets—can cause problems in some people.**	371 (85.1)	146 (82.0)	255 (77.0)	300 (71.8)	279 (68.2)	331 (67.1)	65 (73.0)	1747 (74.2)
**6.** **The used dose needs to be reduced slowly over a few weeks or months before stopping Corticosteroids if you have been taking them for a long time.**	357 (81.9)	144 (80.9)	233 (70.4)	296 (70.8)	286 (69.9)	321 (65.1)	67 (75.3)	1704 (72.4)
**Potential side effects of long-term treatment of Corticosteroids are:**								
**7.** **Skin acne**	365 (83.7)	145 (81.5)	252 (76.1)	281 (67.2)	260 (63.6)	342 (69.4)	64 (71.9)	1709 (72.6)
**8.** **Weight gain**	278 (63.8)	119 (66.9)	180 (54.4)	172 (41.1)	197 (48.2)	296 (60.0)	46 (51.7)	1288 (54.7)
**9.** **Thinned skin that bruises easily**	297 (68.1)	122 (68.5)	182 (55.0)	202 (48.3)	237 (57.9)	290 (58.8)	53 (59.6)	1383 (58.8)
**10.** **Increased risk of infections**	326 (74.8)	136 (76.4)	178 (53.8)	192 (45.9)	232 (56.7)	248 (50.3)	57 (64.0)	1369 (58.2)
**11.** **Mood changes**	284 (65.1)	113 (63.5)	183 (55.3)	219 (52.4)	218 (53.3)	207 (42.0)	54 (60.7)	1278 (54.3)
**12.** **High blood glucose**	299 (68.6)	131 (73.6)	173 (52.3)	156 (37.3)	178 (43.5)	202 (41.0)	52 (58.4)	1191 (50.6)
**13.** **High blood pressure**	302 (69.3)	125 (70.2)	162 (48.9)	162 (38.8)	198 (48.4)	203 (41.2)	52 (58.4)	1204 (51.1)
**14.** **Osteoporosis**	278 (63.8)	124 (69.7)	193 (58.3)	194 (46.4)	197 (48.2)	311 (63.1)	56 (62.9)	1353 (57.5)
**Knowledge score, Mean (STD) ***	**10.5 (3.7)**	**10.5 (3.9)**	**8.6 (4.2)**	**7.5 (4.1)**	**8.1 (4.1)**	**8.0 (5.3)**	**9.1 (4.3)**	**8.7 (4.5)**

Note: each percentage was calculated based on the frequency of the participated country. * Knowledge score has been calculated form the participants answers as the correct answer scored 1 and the incorrect answer scored 0. The total score is 14 points.

**Table 5 healthcare-11-00255-t005:** The mean scores of corticophobia among the study participants from different countries.

Statement	Residential Country, Mean (STD)							
	Egypt	Iraq	Jordan	Saudi Arabia	Sudan	Syria	Others	Total
○ **I am afraid of the weight gains due to Corticosteroids use**	4.3 (0.9)	3.6 (1.1)	3.8 (1.1)	3.9 (1.1)	4.2 (1.0)	3.9 (1.1)	3.9 (1.1)	4.0 (1.1)
○ **I am afraid form the possible increase in blood sugar due to Corticosteroids use**	4.1 (0.9)	4.5 (0.9)	4.0 (0.9)	3.9 (1.0)	4.1 (0.9)	4.1 (0.9)	3.9 (1.0)	4.1 (0.9)
○ **I am afraid form the possible increase in blood pressure due to Corticosteroids use**	4.1 (0.9)	4.3 (0.9)	3.9 (0.9)	3.8 (1.0)	4.1 (0.9)	4.0 (0.9)	3.7 (1.0)	4.0 (1.0)
○ **I am afraid form the possible osteoporosis due to Corticosteroids use**	4.0 (1.0)	4.2 (1.0)	3.9 (1.0)	3.9 (1.0)	4.1 (0.9)	4.0 (1.0)	3.7 (1.0)	4.0 (1.0)
○ **I am afraid that my body will depend on or get addicted to Corticosteroids use and can’t live without it**	4.0 (1.1)	4.2 (1.0)	3.8 (1.1)	3.8 (1.1)	4.1 (1.0)	4.1 (1.0)	3.7 (1.1)	4.0 (1.1)
○ **I am afraid form oral and injectable Corticosteroids more than the topical products**	4.0 (1.1)	4.1 (1.1)	3.8 (1.0)	3.9 (1.1)	3.9 (1.0)	4.0 (1.0)	3.7 (1.2)	3.9 (1.1)
○ **I am afraid of social refusal if they know about my Corticosteroids use**	3.1 (1.3)	3.7 (1.3)	3.2 (1.2)	3.2 (1.3)	3.5 (1.1)	3.7 (1.2)	3.0 (1.2)	3.4 (1.3)
○ **I am afraid form the possible depression of mood swings due to Corticosteroids use**	3.8 (1.0)	3.6 (1.3)	3.6 (1.1)	3.7 (1.1)	4.0 (1.0)	3.8 (1.0)	3.6 (1.1)	3.8 (1.1)
○ **I am afraid of possible unknown/untreatable side effects**	4.1 (1.0)	4.1 (1.0)	3.8 (1.1)	3.9 (1.0)	4.1 (1.0)	3.9 (1.0)	3.7 (1.1)	4.0 (1.0)
○ **I prefer to use traditional therapy rather than Corticosteroids**	3.9 (1.0)	4.1 (1.0)	3.7 (1.0)	3.7 (1.1)	3.8 (1.1)	3.9 (1.0)	3.6 (1.1)	3.8 (1.1)
○ **I prefer to use herbal therapy rather than Corticosteroids**	3.6 (1.1)	3.8 (1.1)	3.7 (1.1)	3.5 (1.2)	3.8 (1.1)	3.8 (1.1)	3.2 (1.2)	3.7 (1.1)
○ **I prefer to use any other medicine, even if expensive, rather than Corticosteroids.**	3.7 (1.1)	3.6 (1.2)	3.7 (1.1)	3.7 (1.1)	3.7 (1.0)	3.8 (1.1)	3.4 (1.1)	3.7 (1.1)
**Overall scores per country, mean (STD)**	**3.9 (0.3)**	**4.0 (0.3)**	**3.8 (0.2)**	**3.7 (0.2)**	**4.0 (0.2)**	**3.9 (0.1)**	**3.6 (0.3)**	**3.9 (0.2)**
**Category of the corticophobia score ***	**High**	**High**	**High**	**High**	**High**	**High**	**High**	**High**

***** Low score on the Likert scale: 1.00–1.66. Moderate score on the Likert scale: 1.67–3.32. High score on the Likert scale: 3.33–5.00.

**Table 6 healthcare-11-00255-t006:** Multivariable logistic regression analysis for the identified independent variables associated with high scores (3.33–5.00) on the Likert scale for corticophobia.

Predictor	*p*-Value	Odds Ratio	95% Confidence Interval	
			Lower Bound	Upper Bound
**Age (Years)**	**0.02**	1.034	1.005	1.063
**Gender (Ref = Male)**				
Female	**0.018**	0.794	0.629	1.571
**Major (Ref = Unemployed)**				
Health sector	0.514	0.787	0.384	1.615
Working in Non-health sector	0.789	1.137	0.444	2.912
Student	0.125	0.257	0.045	1.462
**Educational level (Ref = School level)**				
Bachelor	**0.036**	0.439	0.686	3.02
Diploma	**0.04**	0.894	1.053	9.093
Postgraduates	**0.034**	0.491	0.449	4.954
**COVID-19 Infection (Ref = Yes)**				
No	0.319	1.352	0.747	2.447
**Experience with corticosteroids side effects (Ref = Yes)**	**0.003**	2.126	1.29	3.504
No				
**Knowledge score**	**0.009**	0.841	0.61	1.158

Significance measure at *p* < 0.05 and presented in bold.

## Data Availability

Not applicable.

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
