# Peer review of "Assessment of Knowledge, Perception, Experience and Phobia toward Corticosteroids Use among the General Public in the Era of COVID-19: A Multinational Study"

_healthcare, 2023, doi:10.3390/healthcare11020255_

Round 1
Reviewer 1 Report
It was a pleasure to review the manuscript "healthcare-2095316" titled: "Assessment of knowledge, Perception, Experience and Phobia toward corticosteroids use among the general public in the era of COVID-19: A multinational study" submitted for publication in "Healthcare".
The paper is very pleasant to read, interesting and the topic is important. The paper deserves to be published. However, there are some improvements that must be done to merit publication in this valuable journal. Here are some comments:
1. Line 51, please avoid comma before and.
2. Line 52, it's better to say self-administrated questionnaire (SAQ)
3. Line 53, you don't have to specify the software, please delate it from the abstract.
4. You can write something like "descriptive statistics and logistic regression were computed"
5. Even if your introduction is short, it provides an important source of knowledge and a very good analysis on Corticosteroids. However, your background is lacking, you are studying Corticosteroids in the context of COVID-19. Consequently, you should add at least a pargraph (3-4 sentences) about the disease in general, in your countries, preventive measures, medication....etc. Here are some papers to add:
*https://doi.org/10.1016/j.drup.2021.100794
*https://doi.org/10.3390/healthcare10071341
*https://doi.org/10.1016/j.sapharm.2020.04.001
6. What was the minimum sample size ? I think you have to add the calculation.
7. What about the validation of the questionnaire ?
8. Line 139, please cite all the social media platforms used
9. Table 1, there is a problem in the layout, I can't read everything.
10. Please also specify that percentages (%) are between brackets only one time rather than writing them in all your lines.
11. Line 301-306, please cite these studies from the MENA region, as medication is adressed:
* https://doi.org/10.24926/iip.v12i4.4234
* https://doi.org/10.3390/v14122771
* https://doi.org/10.1002/ptr.7228
12.What about Corticosteroids and vaccination ? this is an interesting perspective
13. I think you have to specify at least the limitations of your study in the end of your discussion section.
I wish the authors good luck.
Author Response
Reviewer 1.
It was a pleasure to review the manuscript "healthcare-2095316" titled: "Assessment of knowledge, Perception, Experience and Phobia toward corticosteroids use among the general public in the era of COVID-19: A multinational study" submitted for publication in "Healthcare".
The paper is very pleasant to read, interesting and the topic is important. The paper deserves to be published. However, there are some improvements that must be done to merit publication in this valuable journal.
R: We would like to thank the reviewer for his valuable input and efforts. We are truly pleased that you liked our work.
Here are some comments:
- Line 51, please avoid comma before and.
R: Thank you for this comment. Amendment was implemented as requested
- Line 52, it's better to say self-administrated questionnaire (SAQ)
R: Thank you for this comment. Amendment was implemented as requested
- Line 53, you don't have to specify the software, please delate it from the abstract.
R: Thank you for this comment. Amendment was implemented as requested
- You can write something like "descriptive statistics and logistic regression were computed"
R: Thank you for this comment. Amendment was implemented as requested
- Even if your introduction is short, it provides an important source of knowledge and a very good analysis on Corticosteroids. However, your background is lacking, you are studying Corticosteroids in the context of COVID-19. Consequently, you should add at least a pargraph (3-4 sentences) about the disease in general, in your countries, preventive measures, medication....etc. Here are some papers to add:
*https://doi.org/10.1016/j.drup.2021.100794
*https://doi.org/10.3390/healthcare10071341
*https://doi.org/10.1016/j.sapharm.2020.04.001
R: Thank you for this comment. General introduction about disease, the situation in Arab countries, preventive measures were added. The recommended articles were cited.
- What was the minimum sample size ? I think you have to add the calculation.
R: Thank you for your comment. Regarding the sample size, we have mentioned the following “based on Tabachnick and Fidell’s recommendation for sample size calculation in analysis, 5–20 subjects per predictor are suggested to be preferable. Based on the num-ber of independent variables levels used in this study (n = 9) and the number of 10 sub-jects per predictor level, a minimum sample size of 90 or higher from each country was considered suitable.”
Ref: 29. Tabachnick, B.G., L.S. Fidell, and J.B. Ullman, Using multivariate statistics. Vol. 5. 2007: pearson Boston, MA.
- What about the validation of the questionnaire ?
R: Thank you for the comment. We have mentioned, “Initially, the tool was developed in English and evaluated by five academics in the pharmacy field for content and face validity. Then, it was translated into Arabic by two independent academic translators using translation and back-translation techniques. For the sake of clarity and understandability of the questionnaire's items among the study-eligible population, it was piloted on 30 participants from various backgrounds (academic and professional), and linguistic refinements were made as needed based on the feedback from the pilot-testing phase.”
- Line 139, please cite all the social media platforms used
R: Thank you for this comment. Done as requested.
- Table 1, there is a problem in the layout, I can't read everything.
R: Thank you for this comment. Amendment was implemented as requested
- Please also specify that percentages (%) are between brackets only one time rather than writing them in all your lines.
R: Thank you for this comment. Amendment was implemented as requested
- Line 301-306, please cite these studies from the MENA region, as medication is adressed:
* https://doi.org/10.24926/iip.v12i4.4234
* https://doi.org/10.3390/v14122771
* https://doi.org/10.1002/ptr.7228
R: Thank you for this comment. Amendment was implemented as requested and references were cited in the discussion section.
12.What about Corticosteroids and vaccination ? this is an interesting perspective
R: Thank you for this comment. We agree with you it is very interesting perspective and we can it in consideration in the future studies.
- I think you have to specify at least the limitations of your study in the end of your discussion section.
R: Thank you for this comment. Amendments has been implemented as requested.
Reviewer 2 Report
The reviewed study needs improvement:
1. the statistical section is too undeveloped and it is difficult to know from the results alone whether the statistical bases were used correctly
2. graphically, especially table 1, is not neatly done
3. the assumptions of the study are not sufficiently explained
4. the results of the work concern patients from one cultural area, which does not translate to other areas of the world and has no global significance
5. the discussion does not exhaustively answer the essential theses
Author Response
Reviewer2
The reviewed study needs improvement:
R: We would like to thank the reviewer for his valuable input and efforts. We are truly pleased that you liked our work.
- the statistical section is too undeveloped and it is difficult to know from the results alone whether the statistical bases were used correctly
R: Thank you for this comment. This the first study of its kind and the used statistical analysis were chosen by the statistician according the research question.
- graphically, especially table 1, is not neatly done
R: Thank you for this comment. Amendments has been implemented as requested.
- the assumptions of the study are not sufficiently explained
R: Thank you for this comment. This issue has been modified in the introduction section.
- the results of the work concern patients from one cultural area, which does not translate to other areas of the world and has no global significance
R: Thank you for this comment. This the first study of its kind and we believe it will have a significant impact at global level, Hence, the participating countries represented different socioeconomic levels and health care systems.
- the discussion does not exhaustively answer the essential theses
R: Thank you for this comment. We tried to cover all the main findings of the study to end up with our conclusion. However, some amendments were implemented to improve the quality, hope to match the expectations.
Reviewer 3 Report
Dear authors
Presented paper is well written and should be published. But it should be improved in several details.
Abstract: conclusion that health care workers should provide informations is general knowledge. You should state how knowledge about awarnes or phobia of patients or genetal public is relevant
Introduction: what be the purpose to know public knowledge-this part should be better explained in third paragraph. Please add why the countries are choosen.
Material: binary logistic regression is it used for nonparametric variables. Why Kruskal test. Pearson is for normal distribution. Explain statistic better.
Res and discussion: is prevalences for some questions conected with having covid or being certain gender. Comparison among countries investigated is lacking and what is the aim of choosing these countries.
Conclusion should be more adressing inveatigation: all findings with remark how results could improve anything. Do not use The role of health care providers is crucial in providing essential counselling for patients regarding the rational use of corticosteroids. You have not investugated this. Concentrate on your findings.
Brst of luck
Author Response
Reviewer 3
Dear authors
Presented paper is well written and should be published. But it should be improved in several details.
R: We would like to thank the reviewer for his valuable input and efforts. We are truly pleased that you liked our work.
- Abstract: conclusion that health care workers should provide informations is general knowledge. You should state how knowledge about awarnes or phobia of patients or genetal public is relevant
R: Thank you for this comment. Amendments has been implemented as requested.
- Introduction: what be the purpose to know public knowledge-this part should be better explained in third paragraph. Please add why the countries are choosen.
R: Thank you for this comment. Reasons to select these countries, and the reason for investigation of public knowledge to corticosteroids were further elucidated in the revised manuscript.
- Material: binary logistic regression is it used for nonparametric variables. Why Kruskal test. Pearson is for normal distribution. Explain statistic better.
R: Thank you for this comment. According to the statistical references “The KruskalWallis test is the non-parametric analogue of a one-way anova, which does not make assumptions about normality” . As well, Pearson's correlation does NOT assume normality.
https://www.tandfonline.com/doi/abs/10.1080/09720510.2012.10701623?journalCode=tsms20
https://stats.stackexchange.com/questions/3730/pearsons-or-spearmans-correlation-with-non-normal-data
- Res and discussion: is prevalences for some questions conected with having covid or being certain gender. Comparison among countries investigated is lacking and what is the aim of choosing these countries.
R: Thank you for this comment. Regression results at 95% confidence interval revealed that a high Corticophobia score was positively correlated with female gender, age, and history of COVID-19 in-fection. Regrading the participating countries, they represented different socioeconomic levels and health care systems.
- Conclusion should be more addressing investigation: all findings with remark how results could improve anything. Do not use The role of health care providers is crucial in providing essential counselling for patients regarding the rational use of corticosteroids. You have not investugated this. Concentrate on your findings.
R: Thank you for this comment. Amendments has been implemented as requested.
Round 2
Reviewer 1 Report
Dear authors,
I would like to thank you for the efforts you put in this revision. You have taken into consideration almost all my comments. In addition, you provided detailed responses to reviewers.
I think the manuscript is now suitable for publication.
With best regards.
Author Response
Thank you very much
Appreciated
Reviewer 2 Report
The authors have slightly improved the readability of the manuscript and made significant changes to all parts of the paper, which has had a positive impact on the perception of the whole. The description of the statistical results obtained (selection of individual elements of the logistic analysis model. In addition, the authors should review the language style in the manuscript.
Author Response
Thank you very much for the feedback.
After an extensive literature review, we chose the sociodemographic predictors, which were used for the study analysis. All the predictors were analyzed using Logistic regression analyses for the categorical predictors , and multiple linear regression was used to examine the continuous variables (i.e., age and knowledge score). In the logistic regression analysis, variables that were independently associated with corticophobia were identified. All the pre-requisite assumptions for the regression models were checked and the variables were selected after checking their independence, where Pearson's correlation coefficient (r) less than 0.9 indicates the absence of multi-collinearity between the independent variables in regression analysis. Statistical significance was considered at a p-value ≤ 0.05.
Reviewer 3 Report
Dear authors,
I beleive that this work is improved and should be published.
Best of regards
Author Response
Thank you very much
Appreciated